# Dopamine: The Neuromodulator of Long-Term Synaptic Plasticity, Reward and Movement Control

**DOI:** 10.3390/cells10040735

**Published:** 2021-03-26

**Authors:** Luisa Speranza, Umberto di Porzio, Davide Viggiano, Antonio de Donato, Floriana Volpicelli

**Affiliations:** 1Dominick P. Purpura Department of Neuroscience, Albert Einstein College of Medicine, 1300 Morris Park Avenue, Bronx, NY 10461, USA; luisa.speranza@einsteinmed.org; 2Institute of Genetics and Biophysics “Adriano Buzzati Traverso”, CNR, 80131 Naples, Italy; 3Department of Translational Medical Sciences, Genetic Research Institute “Gaetano Salvatore”, University of Campania “L. Vanvitelli”, IT and Biogem S.c.a.r.l., 83031 Ariano Irpino, Italy; davide.viggiano@unicampania.it (D.V.); antonio.dedonato@unicampania.it (A.d.D.); 4Department of Pharmacy, School of Medicine and Surgery, University of Naples Federico II, 80131 Naples, Italy; floriana.volpicelli@unina.it

**Keywords:** dopamine, reward, motor control, synaptic plasticity, LTP, LTD, schizophrenia, Parkinson’s disease, vascular dementia

## Abstract

Dopamine (DA) is a key neurotransmitter involved in multiple physiological functions including motor control, modulation of affective and emotional states, reward mechanisms, reinforcement of behavior, and selected higher cognitive functions. Dysfunction in dopaminergic transmission is recognized as a core alteration in several devastating neurological and psychiatric disorders, including Parkinson’s disease (PD), schizophrenia, bipolar disorder, attention deficit hyperactivity disorder (ADHD) and addiction. Here we will discuss the current insights on the role of DA in motor control and reward learning mechanisms and its involvement in the modulation of synaptic dynamics through different pathways. In particular, we will consider the role of DA as neuromodulator of two forms of synaptic plasticity, known as long-term potentiation (LTP) and long-term depression (LTD) in several cortical and subcortical areas. Finally, we will delineate how the effect of DA on dendritic spines places this molecule at the interface between the motor and the cognitive systems. Specifically, we will be focusing on PD, vascular dementia, and schizophrenia.

## 1. Introduction

In the mammalian brain, the dopaminergic (DAergic) systems play a central role in controlling movement, hormone release, emotional balance, reward, odor discrimination and vision.

DAergic neurons are anatomically and functionally heterogeneous, located in the telencephalon, [dispersed within the glomerular layer of the olfactory blub (field A16; [1]) and the amacrine cell population of the retina (field A17; [2]), in the diencephalon where it negatively regulates the production of prolactin (hypothalamic arcuate nucleus, area A12, [3] and sub-parafascicular thalamic nucleus, area A11), which innervate the superior olivary complex and the inferior colliculus in the brain stem where it is supposed to regulate auditory processing (area 13, [4,5]). The most numerous contingent of DAergic neurons, about 70%, resides in the ventral midbrain (mDAergic) to form three distinct nuclei, the substantia nigra (SN, A9), divided into *pars reticulata* (SNpr) and *pars compacta* (SNpc), the ventral tegmental area (VTA, A10) and the retrorubric nucleus (A8).

The development of mDAergic neurons in vivo is a highly coordinated and complex process involving events ranging from neurulation, proliferation and differentiation of progenitor cells to migration, formation of synapse and neural circuits. The external signals such as morphogens and growth factors, activation of specific gene cascades and cellular interactions involved in the specification, differentiation and maturation of DAergic function have been discussed elsewhere (for extensive reviews see [6,7,8,9,10,11,12,13,14]).

Impaired mesencephalic DA neurotransmission is involved in movement disorders, as well as selected psychotic syndromes. The alteration of motor control as a specific syndrome related to DA was described in 1817, by James Parkinson (1755–1824) in his milestones monograph “Essay on the Shaking Palsy” in London. However the disorder was known since antiquity and already treated with the seeds of a legume plant (*Mucuna pruriens*) that contains the therapeutic levels of what is now known as levodopa, used in Parkinson’s disease (PD) treatment.

### 1.1. What Is Dopamine?

DA, also known as 3,4-dihydroxytyramine, is a neurotransmitter produced by DAergic neurons in the brain. DA is synthesized by the tyrosine hydroxylase enzyme, which, by adding a hydroxyl group, transforms tyrosine into L-DOPA, which in turn is decarboxylated into DA. The synaptic vesicle protein VMAT2 (vesicular monoamine transporter 2) carries DA into the vesicles, from which it is released into the synaptic cleft and binds to DARs. DA binds to five receptor subtypes: D1–D5, which are members of the G protein-coupled receptor family (GPCR), classified into two major subclasses: D1R-like and D2R-like receptor families. D1R-like receptors (D1R and D5R) generally couple to the Gs/olf proteins that stimulate adenylate cyclase (AC), the enzyme that converts adenosine triphosphate (ATP) into cyclic adenosine monophosphate (cAMP) and consequently increase cAMP production. The latter activates protein kinase A (PKA), which in turn phosphorylates c-AMP response element-binding protein (CREB), which is translocated into the nucleus activating CREB-dependent transcription of genes involved in synaptic plasticity. D1R modulate different ion channels, including voltage-activated Na^+^, K^+^, Ca^2+^ channels and G-protein gated inwardly rectifying K^+^ (GIRK) channel [15,16,17,18].

Conversely, D2R-like receptors (D2R, D3R and D4R) by coupling to Gi/o proteins, induce inhibition of AC and PKA-dependent pathways, as well as activation of GIRK and closure of voltage-activated Ca^2+^ channels [19]. DA receptors can act as monomers or as dimeric and/or oligomeric complexes by association of different subtypes, either alone or with other GPCRs and ligand-gated channels. D1R-D2R, D2R-D4R, D1R-D3R, D2R-D3R and D2R-D5R exist as homodimers. Oligomeric complexes contain DA receptors associated to the adenosine A1 and A2, serotoninergic 5-HT2A, histaminergic H3, glutamatergic mGlu5 and NMDA receptors [20]. Dimeric/oligomeric complexes display pharmacological and functional properties distinct from their constituent receptors.

DA receptors can also activate G proteins independent mechanisms. The latter is mediated by the multifunctional adaptor protein arrestin, which binds DA receptors phosphorylated by GPCR kinases (GRKs; [21]) and recruits several proteins, including Akt, GSK-3, MAPK, c-Src, Mdm2 and N-ethylmaleimide-sensitive factor [22]. Arrestin binding to active phosphorylated receptors stops further G protein activation and promotes receptor endocytosis. There are seven GRKs in mammals: GRK2, GRK3, GRK4, GRK5 and GRK6 regulate D1R and D2R [23,24,25,26], whereas GRK4 controls D3R [25].

The striatum expresses GRKs 2, 3, 5, and 6 with different expression levels and cellular and subcellular distribution [27].

Therefore, DA is either excitatory or inhibitory depending on which receptors are found on the surface of the target neuron’s membrane and how these neurons respond to the increase or decrease in cAMP concentration. Thus, different functional roles of DA are to be expected that depend on receptor subtype, cell type, synaptic properties, and interactions with other transmitters.

### 1.2. The Mesencephalic Dopaminergic System

Since the middle of the last century it has been understood that the DAergic system has also a fundamental importance in behavioral control and that its dysfunctions generate serious neurological and psychiatric diseases. Its involvement in the orchestration of the neural mechanisms of learning and memory to the pursuit of rewards and the clues behind them is essential for survival behaviors [28]. The stimuli that produce motivation and reward (sex, food, water, drugs, listening to music), increase the release of DA. DA neurotransmission plays a central role also in working memory, the ability to retain information for short periods of time, by changing the density of D1 receptors in the prefrontal cortex (PFC) [29].

In the mammalian central nervous system, three large neural circuits are controlled by SN and VTA mDAergic neurons: the nigrostriatal, the mesolimbic, and the mesocortical pathways (Figure 1).

The nigrostriatal pathway originates in the SNpc and sends its projections to the caudate and putamen nuclei of dorsal striatum. It plays a primary role in controlling motor function and learning motor skills. DA deficiency produces movement disorders as in PD: stiffness, tremors, slowness of movement. While the excess of DA in this area, causes hyperkinetic disorders, such as chorea (involuntary, sudden movements, etc.) or tics.

The mesolimbic pathway originates in the VTA, which sends DAergic axons to the prefrontal cortex, nucleus accumbens (nAc), amygdala, cingulate gyrus, hippocampus, and piriformis complex of the olfactory bulb and the insular cortex [30]. DAergic innervation of the amygdala and cingulate gyrus is highly involved in the formation and processing of emotions. In the hippocampus, the presence of DA endings is associated with learning, working memory, and long-term memory formation. DA binding the DA receptors (DAR) present in the nAc (ventral striatum) and the PFC, causes excitement and influencing motivation and desire for rewarding stimuli, and perception of pleasure and in addictions (Figure 1). It is important to understand the DAergic circuit linked to the gratification mechanism, essential for the motivation and behavioral choice system in mammals. It is an “experience-dependent” learning structure that models motivations to obtain positive hedonistic responses, particularly significant from an evolutionary point of view as survival and reproduction (see below).

In turn, the mesocortical pathway affects decisions and actions by connecting to the PFC, which is responsible for executive skills, planning, and decision making. DA deficiency in this area (as in schizophrenia) reduces reactions to external stimuli, or as in ADHD it causes deficit of attention.

## 2. Synaptic Plasticity: Homosynaptic and Heterosynaptic Theories

The plasticity of the nervous system can be defined as its ability to adapt in response to experience; this property is linked to changes of existing synapses, and to the formation of new synapses, as well as changes of dendritic spines. It is this structural plasticity that sustains lasting changes in the brain with learning and experience. It is this plastic property that gives to the mammalian brain its fascinating ability of elaborating and storing information in a highly organized neuronal network [31]. Synaptic connections between neurons can be modified to respond to changes in the environment, involving axonal and dendritic arborization and pruning, an increase in spine density, and synaptogenesis [32].

The various forms of synaptic plasticity can be collected in three broad types, in accordance to the activity patterns required for the induction, distinct functions served in learning system and different computational roles.

The first is the *homosynaptic plasticity (or Hebbian activity-dependent)*, which requires presynaptic activation of the synapse for the induction. By definition, it occurs only at the synapse that was directly involved in activation of a cell during the induction [33]. The first to propose that the strength of the connection between two neurons is increased for a long period of time when the firing of the pre- and post-synaptic neuron are closely correlated in time was Donald Hebb. Subsequently, this synaptic strengthening has been termed associative because it associates the firing of a postsynaptic neuron with that of a presynaptic neuron [34]. In addition, Hebb implied also that the synaptic strengthening is input-specific: when two neurons fire together their synapse is strengthened but other synapses on either neurons remain unchanged. This form of plasticity underlies a multitude of phenomena in the nervous system—for example, the refinement of connectivity during development (“neurons that fire together wire together”), extraction of causal relations between events in the environment in classical conditioning (or Pavlovian) and other types of associative learning as well as motor learning [33].

The second form is the *heterosynaptic plasticity,* which is not limited to active synapses, but can be induced at synapses that were not active during the induction of homosynaptic plasticity [35]. Since behavioral learning processes such as classical conditioning result from the consequences of one stimulus input on another, Kandel and Tauc proposed the second heterosynaptic rule for strengthening synaptic connections, underlying that a synapse could be strengthened or weakened without a requirement of activity of either the pre- or postsynaptic neurons as a result of the firing of a third, modulatory interneuron [36]. They also suggested that this heterosynaptic modulation could have one of two forms: non-associative or associative. The non-associative form is purely heterosynaptic, whereas associative, activity-dependent heterosynaptic modulation combines features of homosynaptic and heterosynaptic mechanisms.

The third form of plasticity is the homeostatic synaptic scaling characterized by severe and extended changes of activity aiming at the maintenance of the activity levels in an appropriate homeostatic range [37]. Therefore, increased circuit activity led to a decrease in excitatory synapse-wide strength; contrariwise a decrease in the circuit’s activity operate in an increase excitatory synapse-wide [38]. Since homeostatic synaptic scaling is triggered by overall activity level independently which synapse contributed to the induction and changes the weights of all synapses of all cell proportionally, may include changes of both homosynaptic (those active) and heterosynaptic (those not active) inputs [33].

A part from its role as an important neuromodulator involved in motivation and stimulus-reward learning process, and its ability to modulate synaptic plasticity [39], DA can also affect the synaptic strength of neuronal circuits acting on neuronal populations that induce LTP and LTD [40].

As previously described, DA exerts its action through D1-like (D1 and D5) and D2-like (D2, D3, D4) receptors, resulting in increased and decreased cAMP levels respectively. Since D1-like and D2-like receptors act in an opposite way, it is believed that DA can facilitate and depress the synaptic plasticity; usually D1R seems to influence plasticity processes causing disinhibition [41] and modulating NMDAR signaling [42,43,44] (Figure 2).

### 2.1. Dopamine Triggers Heterosynaptic Plasticity in the Hippocampus and Regulates Cognitive Processes

Together with its extensive role in motor control, reward and motivational processes, a large body of new evidence indicates an essential role of DA in learning and memory, and in particular in memory-related neuroplasticity processes. Dopamine is a critical modulator of hippocampal-dependent mnemonic processes, acting differently on various aspects of memory and cognition [45,46,47,48,49].

The real source of DAergic innervation of the hippocampus is still debated. The VTA was proposed as the source for DA to the hippocampus [50,51,52], but several recent findings support the idea that an additional source of DA in the hippocampus come from the locus coeruleus (LC) [46,49,53,54]. Although the source of DA arriving at the hippocampus is debated, the data relative to the distribution of dopamine receptor subtypes in the hippocampus is convincing. In the dorsal hippocampus, dopamine receptors D1-like are prominently expressed in granule cells of the dentate gyrus and are less dense in the CA3 and CA1 area [55]; D2Rs are mainly found in hilar mossy cells [55]. D1/D2 heterodimers are present in the hilar region of the dorsal hippocampus. The ventral hippocampus plays a pivotal role in addiction and other dopamine-dependent psychiatric disorders [56,57]. D1Rs were detected in granule cells of the dentate gyrus, while D2Rs are expressed in the hilus [55]. Little is known about the distribution of subfamily members of the D2-like receptor family.

Thus, physiological and behavioral evidence support that DA receptor signaling influences hippocampal function. Hippocampal plasticity, in terms of LTP and LTD, has been involved in both spatial memory formation and novelty acquisition. The early phase of LTP is governed by Hebbian properties; for the induction of LTP is required a presynaptic input and a strong postsynaptic depolarization [58]. These rules are not sufficient for the persistence/maintenance of the LTP that requires an additional non-local signal. For novel information and motivational events (reward) this signal at hippocampal CA1 synapses is mediated by DA [59]. DA induce the protein synthesis required for the late phase of LTP within dendrites of hippocampal neurons. Experimental evidence suggest that D1 receptor and BDNF mediated pathways interact in the activation of ERK 1/2 [Mitogen activated kinase (MAPK)] [60]. Active ERK1/2 can then induce nuclear transcription via CREB, regulate translation (for example activating the translational initiation factor eIF4E), and stimulate ribosomal function.

Hippocampal late LTP is blocked by a dominant-negative mutation in MEK1, the kinase that directly phosphorylates ERK 1/2 [59]. This form of plasticity in which in addition to the two Hebbian-factors is required a third signal, DA, is termed heterosynaptic modulatory-input dependent [36].

Additional studies in rodent using pharmacological manipulation of hippocampal D1/D5 dopamine receptors provided sufficient evidence that the presumed source of hippocampal DA, coming from the VTA, contributes to memory encoding [61], and also is necessary to convert short-term memory to protein synthesis-dependent long-term memory [62,63,64,65,66].

As mentioned above, new and exciting insights have demonstrated that noradrenergic neurons in the LC corelease noradrenaline and DA in the hippocampus [67].

In particular, Kempadoo and colleagues [46] by optogenetic studies showed that the release of DA from the LC into the dorsal hippocampus enhanced selective attention and spatial object recognition via the dopamine D1/D5 receptors.

Takeuchi et al. [49] reported that LC-TH^+^ neurons project to the hippocampus and optogenetic activation of D1/D5 receptor is involved in novelty-associated memory.

More recently, Wagatsuma and colleagues [68] showed that the neuromodulator input from LC to CA3 but not to CA1 or to the dentate gyrus is crucial for the formation of a persistent memory in the hippocampus. These new findings have established that DAergic signaling from LC is an important source to enhance memory of novelty. Although DAergic VTA and LC projections in the dorsal hippocampus promote memory consolidation, to date it is not clear the reason for the existence of these two separate signals projecting to the hippocampus. Only recently, Duszkiewicz et al. [45] proposed that VTA is activate by experiences related to the past and promote semantic memory formation via memory consolidation. By contrast, the LC is activated by experiences that have only a minimal relationship to the past. The latter causes a consolidation of hippocampal memory and vivid and lifelong episodic memories.

In conclusion, the emerging scenario of long-term regulation of synaptic plasticity in hippocampal circuits includes different types of plasticity regulated by diverse biochemical signaling, in this both the homo- and hetero- plasticity are embraced.

The Hebbian mechanisms may be used primarily for the learning or the short-term memory, but they are not sufficient to support the signaling pathways required for the synaptic growth and maintenance; in contrast, the heterosynaptic mechanisms can recruit long-term memory mechanisms that lead to transcription and to the growth of the new synaptic connections [36].

Thus, to understand the role of DA in the hippocampus is critical for a range of functions from spatial learning to the stable long-term memory.

Deficits in DA are underlying psychiatric disorders, such as schizophrenia [69]. Alteration of DAergic signaling in the hippocampus has also been described in Huntington’s diseases (HD) [70], PD [71], ADHD [72,73], and Alzheimer’s diseases [74].

### 2.2. Dopamine Modulate the Spike Timing Dependent Plasticity (STDP) in Cortical Circuits: A Mechanism of Hebbian Plasticity

Synapses in cerebral cortex are highly dynamic and in part regulated by neuromodulators that act through the gating of plasticity and the up-regulation of neuronal activity.

The neuromodulation of cortical circuits is crucial for working memory [75,76,77], attention [78,79], and PFC depends on flexible behavior. 

Among the neuromodulators, DA modulates spike-timing-dependent plasticity (STDP) in juvenile rodent cortical neurons [80]. 

STDP is a form of Hebbian learning and memory regulated by temporal pairing between the spikes of pre and post-synaptic neurons. During STDP, repeated presynaptic spike arrival few milliseconds before postsynaptic action potentials evokes a LTP at the synapse; instead, repeated presynaptic spike arrival after postsynaptic spikes drives to LTD [61]. 

In this form of plasticity, DA has an important modulatory role to expand the time window for detecting coincident spiking in the pre- and postsynaptic neurons, and in this way prompt to induce the t-LTP in rodent neocortical neurons [81].

### 2.3. Dopamine and Reward System

Both mDAergic neurons of VTA and SN project to PFC via the mesocortical pathway, the latter is separated into two parallel systems [82]: the first is an evolutionarily older system originating from VTA and innervating the anterior cingulate cortex (Brodmann area 24) and medial frontal areas (areas 14 and 32); the second, developed in the primates, originates from dorsolateral and lateral SN to project to the evolutionarily novel and granular dorsal and lateral areas of the PFC (areas 12/47, 9/46, and 9) [82,83,84]. VTA neurons send their diffuse projections also to limbic regions, including the nAc and inform the organism, whether an environmental stimulus (natural reward, drug of abuse, stress) is rewarding or aversive.

The reward system has been the subject of extensive studies. Any stimulus, object, event, activity, or situation that is capable of generating happiness and positive learning (positive reinforcement) can be considered a rewarding stimulus. In scientific terms, reward has three additional functions: (i) to induce approach behavior (whereas punishments induce withdrawal); (ii) to elicit movements towards the desired object, this is a factor to be considered in economic choices; (iii) to arouse emotions, such as pleasure, disgust, pain and fear, feelings that have been tested in animals. Their underlying brain processes have been quantitatively assessed using specific behavioral tasks.

The hedonia and anhedonia hypothesis of DA was postulated in the 1978 by Wise, who demonstrated that DA increase results in the subjective pleasure associated with positive rewards, while the anhedonia is associated with a reduction in DA levels [85]. A critical contribution to the understanding of the role of the DA system in reward processes comes from the works of Wolfram Schultz [28,86,87,88]. The latter initially interpreted the control of phasic firing in VTA as supporting “behavioral activation” [86] and “motivational arousal” [89]. Then, in the ‘90s, medial VTA neurons showed phasic bursts following reward-predicting cues and suppressed activity following punishment-predictive cues, thus the phasic DA bursts were associated to reward prediction error (RPE), that enables reinforcement learning [90].

A RPE is the difference between a reward that is being received and the reward that is predicted to be received. A RPE can be quantified in any useful objective or subjective unit of reward, such as pounds sterling or milliliters of juice or economic utility. Applied to reward, a prediction error will teach that a reward is different from the expected one and that is possible to improve the predictions or correct the behavior.

The advent of optogenetic manipulations in the early 2000s brought a revolution to investigations of DA function. In particular, an important study by Steinberg et al. found that RPE signaling by VTA DAergic neurons is sufficient to support new cue-reward learning and modify previously learned cue-reward associations [91,92]. Recently, Maes et al., using inhibitory optogenetics to prevent cue-evoked DA signals found that the DA neuron activity observed in response to a reward-predictive cue is a prediction error, not a signal about the value of the cue [93]. Experiments using excitatory optogenetics to stimulate DA neuron activity published [94] confirmed that DA stimulation supports associative learning on antecedent cues without evidence of a cached value being ascribed to the cue. Additionally, Morrens et al. [95] showed that novel, but not familiar, cues evoke DA release and that if DA release is inhibited during a novel cue, learning about that cue is impaired. This is in accordance with the idea that DA acts not only as a prediction error for reward values but also as a prediction error for sensory prediction errors [96,97,98].

These studies focused primarily on DAergic neurons in the lateral VTA and suggested that both VTA and SNpc stimulation could support primary reinforcement in self-stimulation paradigms. Furthermore, while Schultz’s original papers on RPE included observations from VTA and SNpc DAergic neurons, recent data [99] demonstrated that optogenetic stimulation of VTA DAergic neurons at the time of reward can unblock learning, while optogenetic stimulation of nearby SNpc DAergic neurons cannot. The lateral portion of the SN, which project to the caudal tail of the striatum, receives a distinct inputs compared to VTA and SNpc, and respond to salient, novel stimuli [100,101,102].

In summary, during RPE DA neurons respond with short, phasic bursts of activity when animals are presented with appetitive stimuli, such as food; then, once the animals have learned to associate a previously irrelevant stimulus with a reward, DA neurons shift their phasic activation from the time of reward delivery to the time of presentation of this predictive cue; finally, unexpected omission of reward leads to a suppression of DAergic neurons.

More data are accumulating that show DAergic neuron heterogeneity in relation to RPE theory [103,104]. Recent transcriptomics and bioinformatics studies of single-cell have enabled the detection of coordinated gene expression profiles within individual cells [105]. Recently molecularly distinct populations of mDAergic neurons have been identified, but their location and molecular signature are only partially overlapping (reviewed by [106]). In particular, Kramer et al. demonstrates that during the postnatal development the expression of Neuronal differentiation 6 (Neurod6) and Gastrin Releasing Peptide (Grp) identifies a population of ventromedial VTA DAergic neurons that sends projections to the medial shell of the nAc [107]. In addition, a recent study genetically isolated VTA DAergic neuron subtypes by the expression of the neuropeptidergic markers Crhr1 (corticotropin-releasing hormone receptor 1) and Cck (cholecystokinin) [108]. These neurons project to the core and medial shell of the nAc, respectively. Both populations are activated simultaneously by cues, actions, and rewards, and contribute distinctly to reward association and motivation. Their co-activation is required for maximizing reinforcement behavior. In midbrain DA neurons the variations in gene expression can influence the neurotransmitter co-release. In addition to DAergic neurons there are cells within the VTA that release glutamate and γ-aminobutyric acid (GABA) [109]. While activation of glutamate neurons of the VTA has been shown to be rewarding in a dopamine-independent manner [110], activation of glutamate projections from the VTA to nAc alone was found to be aversive [111]. GABA projections from the VTA to the nAc have also been shown to promote reinforcement learning [112]. On the contrary, Heymann et al. observed that glutamate release from *Cck*-Cre-expressing neurons in the nAc shell and GABA release in the nAc core from *Crhr1*-Cre-expressing neurons, the behavioral effects are inhibited by the genetic inactivation of DA release, indicating that DA is the principal neurotransmitter to regulate the observed behavior [108].

In summary, single-cell gene expression studies, even if partially discrepant, direct attention to the molecular heterogeneity of the mDAergic system, modifying the conventional anatomical classifications. How and when DAergic neuron diversity is generated during development remains unknown, further studies are still necessary to fully define DAergic heterogeneity at the molecular level and to align these definitions with behavioral observations.

### 2.4. Dopamine in the Striatum and Movement Control

The striatum receives glutamatergic excitatory inputs from the cortex and thalamic structures and DA inputs from SNpc [113]. These inputs converge in the striatum to establish synapse with medium spiny neurons (MSNs), aspiny GABAergic and large cholinergic interneurons [114]. The 95% of MSNs in the striatum represents the output system and is composed of GABAergic neurons expressing D1, D2, and D3 dopamine receptors. The glutamatergic and DAergic projections converge both onto dendritic spines of the same MSN or on cholinergic interneurons.

The canonical view of the interaction between glutamatergic and DAergic neurotransmission in the striatum hypothesized a dual organization of the striatum and of basal ganglia outputs, which led to direct and indirect pathways, with largely opposing effects on thalamo-cortical activity. The direct MSNs (dMSNS) express predominantly dopamine D1R, in contrast, D2R are expressed in indirect MSNs (iMSNs) [115,116]. The D3 receptor is co-expressed with the D1 receptor in the direct pathway neurons, at least in the rodent [15].

Cortical activation produces a release of glutamate that activates MSNs projecting inhibitory signals to the internal segment of globus pallidus (GPi) and SNpr. MSNs are GABAergic neurons that exert an inhibitory action on GABAergic neurons of the SNpr. The inhibition of GPi/SNpr leads to a disinhibition of thalamic glutamatergic neurons, which project to the cortex. This chain of events facilitates the initiation of locomotor activity [115].

Conversely, in the indirect pathway, the striato-pallidal MSNs, which project to the SNpr via the external segment of the globus pallidus (GPe) and the subthalamic nucleus, inhibits the GABAergic neurons of the GPe, leading to a disinhibition of the glutamatergic neurons of the subthalamic nucleus. The excitatory inputs on the subthalamic nucleus neurons activate the SNpr GABAergic neurons projecting to the thalamus and leading to a reduction of locomotor activity [117].

An imbalance in the activity of the striatal direct and indirect pathway MSNs has also been postulated in many neurodegenerative disorders, including PD and HD.

The functional importance of the direct/indirect pathway model in motor generation and control, even if supported by clinical and experimental findings, is still debated. Over recent years, a lot of cross-talk between these two pathways has been discovered.

In the striatum these two pathways are structurally and functionally intertwined and communicate via the striatal interneurons [118]. In particular, three main subtypes of striatal interneurons are implicated in the feedforward and parallel control of striatal circuits: cholinergic interneurons, NOS-positive interneurons and fast-spiking interneurons.

The striatal neurons receive glutamatergic innervations from the thalamus and the cortex. DAergic terminals, originating from the SNpc, release DA onto MSNs and on the different striatal interneurons.

Cholinergic interneurons, expressing D2 and D1/D5 receptors, respond to DA and release acetylcholine (ACh) acting on both presynaptic glutamatergic terminals and on postsynaptic MSNs [118,119,120]. The endocannabinoid system can be a biochemical cross-talk between direct and indirect pathways. Striatal cholinergic interneurons project to both dMSNs and iMSNs. The heterodimer adenosine 2A receptor (A2A) and D2 receptors on cholinergic interneurons decreases the release of ACh. The decreased levels of ACh on the M1 muscarinic receptors located on dMSNs and iMSNs blocks the L-type calcium current. The increase in intracellular calcium concentration induces endocannabinoid release at the postsynaptic sites of dMSNs and iMSNs. Endocannabinoid acts as retrograde messengers on CB1 cannabinoid receptors located on glutamatergic terminals, blocking the glutamate release. Thus, cholinergic neurons, expressing D2Rs facilitate LTD induction, presumably through the reduction of ACh release and subsequent M1 receptor action [121]. Thus, the cholinergic interneuron represents the cellular substrate for the synaptic cross-talk between the two classes of MSNs. Immunohistochemical characterization of substance P–positive (direct pathway) and A2A receptor–positive (indirect pathway) MSNs, confirmed that D2-dependent LTD is present in both classes of MSNs [122].

Recent findings published by Augustin et al. [123] are in agreement with previous reports, since they demonstrate that deletion of D2Rs in cholinergic interneurons (Chl-Drd2KO) impairs LTD induction in both subtypes of MSNs. Gene targeted deletion of D2R iMSNs and cholinergic interneurons demonstrates that D2 receptors on cholinergic interneurons strongly modulate LTD in all MSNs, while D2Rs on iMSNs can further regulate LTD induction at synapses onto that MSN subtype. LTD induction is restored in the Chl-Drd2KO mice by an M1-selective muscarinic acetylcholine receptor antagonist.

Interestingly, D2R on striatal cholinergic interneurons are involved also in catalexia induced by neuroleptics (D2 antagonists; [124]). MSN express four GRK isoforms, 2, 3, 4, and 5. The cholinergic interneurons are enriched with GRKs 2 and 3, as compared to MSN, whereas the levels of GRKs 5 and 6 do not differ between MSN and interneurons [15]. GRK2 knock-out mice display DAergic dysfunctions and consequently altered movements [125].

The other interneurons involved in direct and indirect interconnections are NOS-positive interneurons and fast-spiking interneurons. NOS-positive interneurons expressing D1/D5 receptors, respond to DA and produce NO that acts as a retrograde messenger and facilitates LTD at the postsynaptic level.

Fast-spiking interneurons, instead, release GABA on MSNs inhibiting the direct and indirect pathways. The induction of either LTP or LTD in MSNs regulates the striatal control on output structures and motor activation/inhibition.

The existence of this cross-talk between direct and indirect pathways is supported from the evidence obtained in non-human primates in which the striatal neurons projecting to either the GPi or the GPe show immunolabeling for both D1 and D2 DA receptors [126].

In PD the severe DA denervation leads to a complete loss of striatal synaptic plasticity of MSNs and alters the physiological activity of striatal interneurons, as well as the neurochemical signals that originate from these cells, causing bradykinesia, freezing, and gait festination

The integration of direct and indirect pathways considers the importance of striatal interneurons in striatal physiology and suggests that all MSNs might either facilitate or inhibit movement depending on the form of synaptic plasticity expressed in a definite moment. Recently we showed that important motor learning processes in cortical and subcortical neural systems, such as the basal ganglia are sustained by spine complexity along dendrites and their remodeling; the latter being a possible general feature associated with the structural plasticity underlying processes such as long-term memory maintenance, reward, motivation, and goal-directed behavior, exerted by the DAergic system [127].

Current data support the idea that a representation of movement in VTA DA neurons is present. Using a “Go-No-Go” task, Syed et al. demonstrated that if an animal do not need to initiate a movement to obtain a reward, the DA release in the nAc is reduced [128].

Engelhard et al., using 2-photon calcium imaging through an implanted lens to record the activity of >300 mDAergic neurons in the VTA during a complex decision-making task demonstrated that DAergic neurons were functionally clustered and the subpopulations of neurons transmitted information about a subset of behavioral variables, in addition to encoding reward [104]. Furthermore, Hughes et al. [129], using in-vivo electrophysiology and optogenetics, identified three populations of VTA DAergic neurons that control the force exerted over time. These populations differ in magnitude, direction, and duration of force used by the animal during motivated behavior. Furthermore, optogenetic stimulation regulates anticipatory licking. Based on these results, it seems possible that DAergic neurons track both movement and RPE often simultaneously in the same cells.

## 3. Dopamine System at the Interface between Motor Control and Cognitive Functions

As discussed above, DA is largely involved in synaptic plasticity. This occurs through the action of DA on dendritic spines. Considering the large territory innervated by midbrain DA terminals, we shall expect that DA modifies several cognitive domains, apart the motor and reward systems. Therefore, here we outline how the effect of DA on dendritic spines places this molecule at the interface between the motor and the cognitive systems. Specifically, we will be focusing on diseases such as PD, vascular dementia, and schizophrenia.

The classical division of our mind into a motor control system and cognitive machinery goes far back into the past. It may even be found in ancient philosophers. While today we accept that emotions and rationality cannot be easily separated (reviewed in the book by Antonio Damasio “Descarte’s Error”), we still have difficulties thinking at an overlap between cognition and motor functions. However, almost all psychiatric conditions and dementias (including Alzheimer’s disease that causes 50–80% of cases, Frontotemporal dementia, and vascular dementia): have subtle modifications in motor control. Conversely, PD, considered a purely motor syndrome, is always accompanied by cognitive dysfunction, as well as other movement disorders, such as HD or Progressive supranuclear palsy.

Indeed, DA depletion in PD is accompanied by altered spine morphology in the cerebral cortex. These changes lend support to cognitive function involvement in this disease [130]. Through specific neuropsychological cognitive tests, it was identified that 93% PD patients had alterations in the main visuospatial, memory, and executive functions [131]. This is due, at least in part, to DA imbalance in PD, as the treatment with L-DOPA can prevent cognitive function decline [132].

However, it should not be forgotten that L-DOPA and DA could also have a neurotoxic effect, leading to depletion of neurons [133]. Furthermore, L-DOPA decreases dendritic spines in the primary motor cortex, and prefrontal cortex [134].

The dopamine-dependent synaptic plasticity emerges also in physiological situations such as aging. A recent study reports that the DAergic innervation of the striatum changes with age, leading to slower cognitive functions [135]. Besides, aging is correlated with a progressive decrease in DAT expression levels in the striatum [136,137,138,139]. The reduction of DAT with aging has several possible interpretations: (i) a decreased concentration of DAT per fiber, thereby increasing the availability of DA at the synaptic level, (ii) a decrease in the number of DA fibers. It is largely accepted that the number of DAergic neurons decreases with age [140]. However, the rate of DA release [141] and the basal DA extracellular level [142] is not modified by aging. Therefore, we should conclude that the decrease of DAT in the striatum is due to both fewer DAergic fibers and a compensatory DAT decrease per axonal terminal. The net effect is a conserved amount of DA in the striatum.

Given this evidence, are DAT blockers beneficial in age-related vascular dementia? DAT blockers (amphetamines, methylphenidate) are often considered “cognitive enhancers” due to their ability to speed up cognitive processing [143]. Therefore, they have been tested in various forms of dementia [144], with improvement in apathy and mood. Unfortunately, a single dose of methylphenidate does not improve cognitive decline [144] or has a minimal effect [145] in the case of vascular dementia. No literature data are available concerning a chronic treatment of DAT blockers on age-dependent vascular dementia.

Finally, DAergic system dysfunction seems to accompany schizophrenia [146], a complex psychiatric condition characterized by hallucinations and disordered thinking. Indeed, L-DOPA can induce psychotic symptoms [147]. Furthermore, drugs acting on the DAergic system, such as antipsychotic drugs (olanzapine) reverse dendritic spine dystrophy observed in schizophrenia [148]. These data strongly support the view that the cognitive effects of DA are mediated by its action on dendritic spines.

The cognitive effects of DA in schizophrenia are mediated by a subtype of D4R DAergic receptors. D4Rs are significantly expressed in the mPFC [149,150] and implicated in gamma oscillations [151]. It has been suggested that cognitive processes are influenced by DA through the modulation of the signal-noise ratio in PFC microcircuits mediated by dendritic spines [151].

Overall, DA mediates both motor functions and cognitive functions, and this stems from the effects of DA on dendritic spines in large brain territories. Therefore, it is not surprising that animal models with genetic alterations of the DAergic machinery show altered exploration of new environments [152] anxiety [153], and memory dysfunction [154].

## 4. Conclusions

The neurotransmitter DA is used by various groups of neurons, but the most consistent DAergic system resides in the midbrain where it constitutes three important nuclei. Midbrain DA has a broad spectrum of action. It is important for excitement, movement, mood, and the execution of activities that require immediate decisions, learning through reward, and the fundamental role of this last aspect for the survival of the species. DA, therefore, influences learning and motivation, and its levels continuously signal how optimal a given situation is for obtaining a reward. This information helps the individual to decide, whether to achieve a goal and helps learn from missing rewards.

Here, we review laboratory and clinical evidence related to this group of neurons for their relevant role in the brain. In particular we highlighted how the brain’s DAergic system forms a series of nerve pathways that moderate the control of behaviors and movement. We summarize how the DAergic system is activated by feelings of reward and how its malfunction can generate addiction to drugs and alcohol, compulsive sexual activity or gambling. We also delineate the well-known role of DAergic innervation forming the extrapyramidal pathway that regulates muscular tone and movement, whose alteration determines PD. Here, we also describe how a series of axons influences cognition in the frontal lobes while another branch activates the limbic system of the temporal lobe, where DA intervenes by increasing the correlation between pleasure and certain behaviors. Finally, we emphasize the important role of DAergic innervation in the cognitive and mnemonic system through innervation in the cortex and the hippocampus.

An accurate DAergic signaling in the neuronal connections is necessary for a correct synaptic plasticity and cognitive functions. Instead, an impaired DAergic modulation of synaptic transmission, is associated with neuropsychiatric disorders such as schizophrenia, ADHD, Obsessive-Compulsive Disorder and Tourette’s syndrome.

## Figures and Tables

**Figure 1 cells-10-00735-f001:**
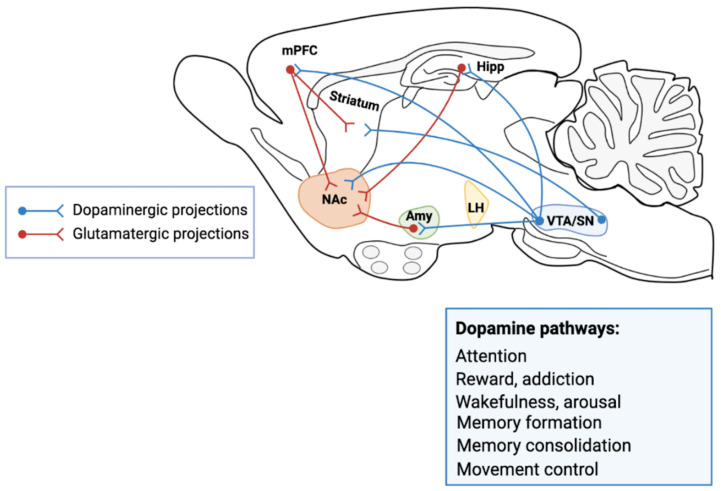
A simplified schematic of the major connections to and from the ventral tegmental area (VTA) and substantia nigra (SN) in the rodent brain. The primary reward circuit includes DAergic projections from the VTA to the nucleus accumbens (nAc), which release dopamine (DA) in response to reward-related stimuli. The nAc receives dense innervation from glutamatergic neurons from the medial prefrontal cortex (mPFC), hippocampus (Hipp) and amygdala (Amy), as well as other regions (Lateral Hypothalamus, LH). The caudate and putamen nuclei of striatum receive dense innervation from zona compacta of the SN and from PFC. The DAergic pathways control attention, addiction, reward, wakefulness, arousal, memory formation and memory consolidation. The illustration was prepared from scratch and created with our original design using Biorender.com.

**Figure 2 cells-10-00735-f002:**
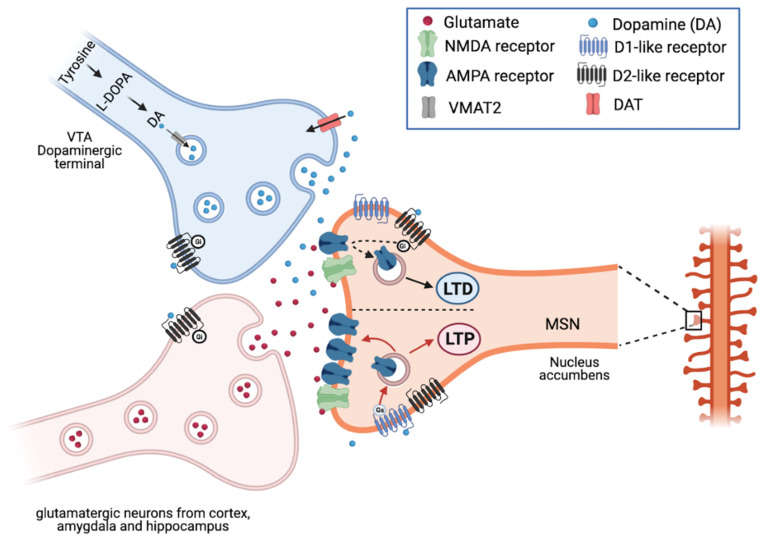
Medium spiny neurons of nucleus accumbens (nAc) receive glutamatergic inputs from medial prefrontal cortex (mPFC) hippocampus and amygdala (Amy) and DAergic inputs from VTA. Glutamate acts on glutamate receptors (AMPA, NMDA), the activation of these receptors is responsible for basal excitatory synaptic transmission and many forms of synaptic plasticity such as Long-Term Potentiation (LTP) and Long-Term Depression (LTD). DA neurons from VTA project to nAc. Dopamine (DA) is synthetized from amino acid tyrosine; the tyrosine hydroxylase, the rate-limiting enzyme of catecholamine synthesis, catalyzes the addition of a hydroxyl group to the meta position of tyrosine, yielding L-DOPA. The latter is rapidly converted to DA by dopa decarboxylase, which is located in the cytoplasm. After synthesis, the vesicular monoamine transporter 2 (VMAT2) transports DA from the cytoplasmic space into synaptic vesicles within presynaptic terminals. Once released, the DA can bind to and activate both presynaptic and postsynaptic DAergic receptors, D1 and D2-like receptors. D1R-like receptors generally couple to the Gs proteins that stimulate adenylate cyclase (AC) and cyclic adenosine monophosphate production. Conversely, D2R-like receptors by coupling to Gi proteins, induce inhibition of AC and PKA-dependent pathways. DA is taken back up into DAergic presynaptic terminals by the DA transporter (DAT). DA can modulate the postsynaptic terminal binding to D1-like receptors can potentiate AMPA and NMDA currents, and stimulate cAMP-dependent signaling. DA binding to D2-like receptors reduce AMPA and NMDA currents and inhibit cAMP-dependent signaling cascades. The illustration was prepared from scratch and created with our original design using Biorender.com.

## Data Availability

Not applicable.

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
