# Peer review of "Dopamine: The Neuromodulator of Long-Term Synaptic Plasticity, Reward and Movement Control"

_cells, 2021, doi:10.3390/cells10040735_

Round 1

Reviewer 1 Report

It is a very interesting review on role of dopamine in long term synaptic plasticity. It is comprensible from basic to specialiced concepts. my minor concern is relate witih the extention of the conclusion it shoul be shortened. 

Author Response

Response to Reviewer 1 Comments

Please find below, in red, detailed point-by-point responses to the reviewers’ comments.

Point 1: It is a very interesting review on role of dopamine in long term synaptic plasticity. It is comprensible from basic to specialized concepts. My minor concern is related with the extention of the conclusion it should be shortened. 

Response 1: We have shortened the conclusions (lines 744-768)

We would like to thank the reviewer for his constructive criticisms and insightful comments that helped us to improve our review.

Reviewer 2 Report

I congratulate you for this extensive review, I just wonder if the images you show are designed by you. If they are from other authors, it is not referenced, if they are yours, they must comment on it.

Author Response

Response to Reviewer 2 Comment

Please find below, in red, detailed point-by-point responses to the reviewers’ comments.

Point 1: I congratulate you for this extensive review, I just wonder if the images you show are designed by you. If they are from other authors, it is not referenced, if they are yours, they must comment on it.

Response 1: The illustration was prepared from scratch and created with our original design using Biorender.com, we added this sentence in the figure legends (lines 214-215 and lines 394-395).

We would like to thank the reviewer for his constructive criticisms and insightful comments that helped us to improve our review.

Reviewer 3 Report

The manuscript from Speranza et al. is a well written compendium of the dopamine role and function in health and disease. The article summarizes the large body of literature about the recent discoveries on DA functions.

Minor:

The authors claim ‘Here we will discuss the current insights on the role of DA in motor control and reward learning…’ but in the paper there is a long excursus in the history of science (i.e. page 2, lines 54-55; page 3, lines 109-110; page 4, lines 165-166; page 5, lines 174-176; etc. etc.) that makes the reading less fluent.

Introduction:

  • dopaminergic systems (DA) please change to dopaminergic (DA) systems

  • ‘Impaired mesencephalic DA neurotransmission is involved in movement disorders, as 46well as selected psychotic syndromes. External signals such as morphogens and growth47factors, activation of specific gene cascades and cellular interactions are involved in the specification and maturation of DA function during neurogenesis and in the differentiation of human and murine embryonic neurons and pluripotent stem cells in vitro’

These sentences are not really linked, please clarify before this points

  • ‘Although the English Thomas Willis (1667) [15]and the Swedish Emanuel Swedenborg 54(1740) indicated a role of the striatum in sensation [16], until the XXth century attention 55prevailed on movement disorders and their roles in the selection, initiation and execution 56of voluntary movement, neglecting its important role as aprediction error signaling 57The alteration of motor control as aspecific syndrome was described in 1817, by58James Parkinson (1755–1824) inhismilestones monograph"Essay on the Shaking Palsy" 59in London. Howeverthedisorderwas known since antiquity and already treated with 60the seeds of a legume plant (Mucuna pruriens) that contains thetherapeutic levels of what61is now known as levodopa, used in PD treatment.’

Although very interesting this historical excursus it’s not very appropriate for introducing the topic (the description of the mesencephalic DA system).

  • PD treatment, acronym not specified before

Maybe the paragraph ‘What is dopamine?’ should be moved before the description of the dopaminergic system since the authors are describing the system before the essential component: the neurotransmitter. Moreover, the title is Dopamine: the neuromodulator of long-term synaptic plastici-2ty, reward and movement control’.

Author Response

Response to Reviewer 3 Comments

Please find below, in red, detailed point-by-point responses to the reviewers’ comments.

The manuscript from Speranza et al. is a well written compendium of the dopamine role and function in health and disease. The article summarizes the large body of literature about the recent discoveries on DA functions.

Minor:

Point 1: The authors claim ‘Here we will discuss the current insights on the role of DA in motor control and reward learning…’ but in the paper there is a long excursus in the history of science (i.e. page 2, lines 54-55; page 3, lines 109-110; page 4, lines 165-166; page 5, lines 174-176; etc. etc.) that makes the reading less fluent.

Response 1: To make the text more fluent as suggested we have deleted some sentences concerning historical references.

Point 2: Introduction ..….dopaminergic systems (DA) please change to dopaminergic (DA) systems

Response 2: We changed the sentence (line 33)

Point 3: ‘Impaired mesencephalic DA neurotransmission is involved in movement disorders, as 46 well as selected psychotic syndromes. External signals such as morphogens and growth 47 factors, activation of specific gene cascades and cellular interactions are involved in the specification and maturation of DA function during neurogenesis and in the differentiation of human and murine embryonic neurons and pluripotent stem cells in vitro’

These sentences are not really linked, please clarify before this points

Response 3: We clarified this point changed the sentence (lines 46-51)

Point 4: ‘Although the English Thomas Willis (1667) [15] and the Swedish Emanuel Swedenborg 54 (1740) indicated a role of the striatum in sensation [16], until the XXth century attention 55 prevailed on movement disorders and their roles in the selection, initiation and execution 56 of voluntary movement, neglecting its important role as a prediction error signaling 57 The alteration of motor control as aspecific syndrome was described in 1817, by 58 James Parkinson (1755–1824) in his milestones monograph "Essay on the Shaking Palsy" 59 in London. However the disorder was known since antiquity and already treated with 60 the seeds of a legume plant (Mucuna pruriens) that contains the therapeutic levels of what61is now known as levodopa, used in PD treatment.’

Although very interesting this historical excursus it’s not very appropriate for introducing the topic (the description of the mesencephalic DA system).

Response 5: A part of the sentence was deleted and another part was moved in the introduction (lines 52-58)

Point 5: PD treatment, acronym not specified before

Response 5: We specified the PD acronym (line 58)

Point 6: Maybe the paragraph ‘What is dopamine?’ should be moved before the description of the dopaminergic system since the authors are describing the system before the essential component: the neurotransmitter. Moreover, the title is Dopamine: the neuromodulator of long-term synaptic plasticity, reward and movement control’.

Response 6: As suggested we moved the paragraph 1.2 entitled “What is dopamine?” before, now is the paragraph 1.1. (lines 62-147)

We would like to thank the reviewer for his constructive criticisms and insightful comments that helped us to improve our review.